# CaMKII Inhibition Attenuates Distinct Gain-of-Function Effects Produced by Mutant Nav1.6 Channels and Reduces Neuronal Excitability

**DOI:** 10.3390/cells11132108

**Published:** 2022-07-04

**Authors:** Agnes S. Zybura, Firoj K. Sahoo, Andy Hudmon, Theodore R. Cummins

**Affiliations:** 1Program in Medical Neuroscience, Paul and Carole Stark Neurosciences Research Institute, Indiana University School of Medicine, Indianapolis, IN 46202, USA; azybura@alumni.iu.edu; 2Department of Medicinal Chemistry and Molecular Pharmacology, College of Pharmacy, Purdue University, West Lafayette, IN 47907, USA; fsahoo@purdue.edu (F.K.S.); ahudmon@purdue.edu (A.H.); 3Biology Department, School of Science, Indiana University-Purdue University Indianapolis, Indianapolis, IN 46202, USA

**Keywords:** CaMKII, Nav1.6, phosphorylation, sodium channel, electrophysiology, post-translational modification (PTM)

## Abstract

Aberrant Nav1.6 activity can induce hyperexcitability associated with epilepsy. Gain-of-function mutations in the *SCN8A* gene encoding Nav1.6 are linked to epilepsy development; however, the molecular mechanisms mediating these changes are remarkably heterogeneous and may involve post-translational regulation of Nav1.6. Because calcium/calmodulin-dependent protein kinase II (CaMKII) is a powerful modulator of Nav1.6 channels, we investigated whether CaMKII modulates disease-linked Nav1.6 mutants. Whole-cell voltage clamp recordings in ND7/23 cells show that CaMKII inhibition of the epilepsy-related mutation R850Q largely recapitulates the effects previously observed for WT Nav1.6. We also characterized a rare missense variant, R639C, located within a regulatory hotspot for CaMKII modulation of Nav1.6. Prediction software algorithms and electrophysiological recordings revealed gain-of-function effects for R639C mutant channel activity, including increased sodium currents and hyperpolarized activation compared to WT Nav1.6. Importantly, the R639C mutation ablates CaMKII phosphorylation at a key regulatory site, T642, and, in contrast to WT and R850Q channels, displays a distinct response to CaMKII inhibition. Computational simulations demonstrate that modeled neurons harboring the R639C or R850Q mutations are hyperexcitable, and simulating the effects of CaMKII inhibition on Nav1.6 activity in modeled neurons differentially reduced hyperexcitability. Acute CaMKII inhibition may represent a promising mechanism to attenuate gain-of-function effects produced by Nav1.6 mutations.

## 1. Introduction

Epilepsy is a group of debilitating neurological disorders characterized by unpredictable seizures, affecting more than 70 million people worldwide [1]. Many pathogenic changes contribute to epilepsy-associated seizure onset, including changes in expression and activity of the voltage-gated sodium channel Nav1.6. Ubiquitously expressed in the nervous system, Nav1.6 displays unique characteristics that make it ideal for controlling electrical impulses in the brain. For example, Nav1.6 is enriched at the axon initial segment (AIS) and nodes of mature neurons [2,3,4], where it orchestrates action potential (AP) initiation and propagation, activates at more negative potentials compared to other neuronal Navs, and can generate large persistent and resurgent currents [5,6,7,8,9], which can amplify neuronal synaptic inputs and mediate repetitive AP firing [10,11,12,13,14]. Therefore, alterations to Nav1.6 activity and expression may profoundly impact neuronal physiology and contribute to pathogenic phenotypes.

More than 150 distinct epilepsy-related mutations have been identified in the *SCN8A* gene that encodes Nav1.6 [15]. These mutations often occur as de novo missense mutations resulting in gain-of-function effects on Nav1.6 activity, including increased sodium currents and premature activation, and they powerfully contribute toward seizure susceptibility [15,16,17]. Indeed, increased Nav1.6 expression has been observed in various seizure models [18,19], while reduced Nav1.6 activity and expression have been found to be neuroprotective and decrease seizure susceptibility [20,21,22,23]. Moreover, the molecular mechanisms and phenotypic outcomes associated with *SCN8A* epilepsies are remarkably heterogeneous, suggesting additional underlying molecular mechanisms beyond mutation.

Navs undergo extensive regulation by post-translational mechanisms and genetic modifiers that can impact channel function and influence excitatory neuronal networks [24,25,26,27,28]. The Ca^2+^/calmodulin-dependent protein kinase II (CaMKII) is a multifunctional Ser/Thr protein kinase ubiquitously expressed throughout the brain and is critical for neuronal excitability [29]. Fluctuations in CaMKII activity are often associated with neuronal diseases, including epilepsy, and can act as a genetic modifier that powerfully regulates ion channels [27,28,30,31,32,33,34]. We previously reported CaMKII-dependent modulation of Nav1.6 properties at two key phosphorylation sites, S561 and T642, and that acutely inhibiting CaMKII results in loss-of-function effects on channel activity [27]. However, it is unknown whether acute CaMKII inhibition may attenuate gain-of-function effects and associated hyperexcitability caused by *SCN8A* epilepsy mutations, or if CaMKII can differentially modulate channel activity between diverse *SCN8A* polymorphisms. 

In this study, we examined the impact of CaMKII modulation on a previously characterized gain-of-function *SCN8A* epilepsy mutant, R850Q [16], using whole-cell patch clamp electrophysiology. This variant is clinically associated with early infantile epileptic encephalopathy (EIEE), as indicated in the ClinVar database, and exhibits aberrant channel properties suggestive of pathogenesis. We also investigated the impact of a rare *SCN8A* variant, R639C, which lies in a hotspot for CaMKII regulation of Nav1.6, and observed gain-of-function effects on channel activity. Although there are fewer clinical descriptions available for R639C, this variant is also associated with EIEE in the ClinVar database. Multiple biochemical approaches, including immobilized and soluble peptide arrays, were used to determine whether the R639C channel retained CaMKII phosphorylation at a key regulatory site, T642, which resides in the Nav1.6 L1 region. Whole-cell voltage clamp electrophysiology was used to determine the impact of CaMKII modulation on R639C activity. Notably, we discovered differential channel modulation with CaMKII between the R850Q and R639C mutant channels. Using computational simulations, we further demonstrate that CaMKII inhibition decreased excitability of modeled Purkinje and cortical pyramidal neurons, indicating that acutely inhibiting CaMKII may differentially attenuate hyperexcitability associated with these mutant channels.

## 2. Materials and Methods

### 2.1. Voltage-Gated Sodium Channel Nav1.6 Expression Construct and Site-Directed Mutagenesis

Codon-optimized human Nav1.6 encoding the amino-acid sequence corresponding to the accession number NP_055006.1 in the NCBI database was designed in-house, synthesized by Genscript (Piscataway, NJ, USA), and subcloned into the pcDNA3.1 mammalian expression vector using KpnI and BamHI restriction sites. This Nav1.6 construct was rendered resistant to tetrodotoxin (TTX) via a Y371S substitution, which does not alter its biophysical properties [35]. The R639C and R850Q point mutations were introduced into wildtype TTX-resistant Nav1.6 cDNA using the QuikChange II XL site-directed mutagenesis kit from Agilent Technologies (Santa Clara, CA, USA) following the manufacturer’s instructions. R639C primers included GCGCAGCGTCAAGTGCAACAGCACCGTGG and CCACGGTGCTGTTGCACTTGACGCTGCGC. R850Q primers included CATTCCGCCTGCAAGTGTTTAAACTGGC and GCCAGTTTAAACACTTGCAGCAGGCGGAATG. Mutant constructs were fully sequenced (ACGT, Inc.) to confirm the correct mutation and absence of off-target mutations in the *SCN8A* gene.

### 2.2. Cell Culture

ND7/23 cells (Sigma Aldrich #92090903) were cultured in DMEM containing 4.5 g/L d-glucose, l-glutamine, and 110 mg/L sodium pyruvate (Thermo Fisher, Waltham, MA, USA) supplemented with 10% fetal bovine serum (FBS, Nucleus Biologicals, San Diego, CA, USA), 10 units/mL penicillin, and 10 µg/mL streptomycin at 37 °C in 5% CO_2_. Cells were passaged at subconfluency (85–90%) by mechanical dissociation. Transient transfections of ND7/23 cells were performed at 80% confluency using Lipofectamine 2000 (0.5 µL/cm^2^, Invitrogen, Waltham, MA, USA) according to the manufacturer’s instructions with either 4 µg of WT or mutant TTX-R Nav1.6 construct and 250 ng of separate soluble EGFP construct, used for positive identification of transfected cells. Accessory subunits were not included in the transfection, as multiple plasmids could increase transfection variability in some cases. Following transfection for 4 h, the cells were washed and passaged at a lower density onto 35 mm cell culture dishes and incubated at 30 °C overnight to facilitate protein surface expression [27]. Transfected cells were identified by robust EGFP expression using a fluorescence microscope, and whole-cell patch clamp recordings were obtained 24–36 h post-transfection. 

### 2.3. CaMKII and Calmodulin Purification

A baculoviral expression system was used to express recombinant human αCaMKII [36] purified with minor modifications [37]. Recombinant calmodulin was expressed in *E. coli* and purified as previously described [37].

### 2.4. Peptide Spots Phosphorylation Assay

Immobilized tiled peptide arrays of the Nav1.6 intracellular L1 region corresponding to amino acids 631–649 were tiled, skipping two residues between consecutive peptides (631–645, 633–647, and 635–649) and spotted onto derivatized cellulose membranes (Intavis AG, Cologne, Germany) as previously described [27,37]. Peptides with the R639C substitution in addition to non-phosphorylatable Ala substitutions at known CaMKII modulatory sites (S641/T642) were also synthesized [27]. Known CaMKII peptide substrates autocamtide-2 (AC2, KKALRRQETVDAL) and GluN2B (RNKLRRQHSYDTFVD) were added to the blot as positive controls to validate signal intensity, in addition to a peptide that is not phosphorylated by CaMKII (N-term of EAAT1, MTKSNGEEPKMGGRM), which provided a negative control. Following synthesis, membranes were processed and subjected to a phosphorylation reaction using human αCaMKII as previously described [27]. Phosphorylated peptides were detected using an Azure Biosystems (Dublin, CA, USA) phospho-imager and quantified using the Azure Biosystems Array system.

### 2.5. Soluble Peptide Synthesis and Purification

The WT Nav1.6 peptide (RSVKRNSTVDCNGVV) and mutant R639C (RSVKCNSTVDCNGVV) peptide were synthesized by solid phase (ProTide^TM^ resin from CEM Corporation, Matthews, NC, USA) Fmoc chemistry using a Liberty Blue™ Automated Microwave Peptide Synthesizer (CEM Corp., Matthews, NC, USA). Peptides were cleaved from the resin using a cleavage cocktail (trifluroacetic acid (94%), 2,2′-(ethylenedioxy) diethanethiol (2.5%), triisopropylsilane (1%), and water (2.5%)), dried under nitrogen, washed with anhydrous diethyl ether, and collected following centrifugation at 4K rpm for 20 min. Subsequently, peptides were resuspended in the presence of reducing agent (10 mM DTT) and immediately purified using a Varian Prostar preparative HPLC system with a Jupiter^®^ 4 μM Proteo 90 A° LC column (Phenomenex, Inc., Torrance, CA, USA). A water and acetonitrile solvent system with 0.1% TFA was used and the peptides were purified using a 10–45% gradient of acetonitrile. Purified peptides were frozen and then lyophilized. Peptide mass was confirmed using MALDI/TOF mass spectrometry at the Analytical Mass Spectrometry facility at Purdue University.

### 2.6. Soluble Peptide Assays

The WT and mutant peptides were dissolved in 20 mM HEPES pH 7.4 with 10 mM DTT and used in our standard kinase assay [27,32,37]. Briefly, each kinase reaction contained 50 mM HEPES pH 7.4, 100 mM NaCl, 10 mM MgCl_2_, 0.5 mM CaCl_2_, 5 μM CaM, 100 μM ATP, 60 μCi/mL [γ-^32^P]-ATP, 100 μM peptide substrates (WT or mutant), and 10 nM αCaMKII. Kinase reactions were carried out at 30 °C for 2 min and spotted onto Whatman Grade P81 ion-exchange chromatography paper (Whatman, GE Healthcare, Piscataway, NJ, USA). The reaction was quenched in 75 mM phosphoric acid (followed by 3 × 5 min washes) to remove unincorporated radioactivity. ^32^P incorporation was quantified using Beckman LS6500 scintillation counter with liquid scintillant (EcoLite^TM^ (+)). The CaMKII substrate peptide, AC2 (KKALRRQETVDAL), was used as a positive control. The kinase reaction for the WT peptide was linear for >4 min at 30 °C, while mutant peptide demonstrated no time-dependent increase in phosphorylation.

### 2.7. Whole-Cell Voltage Clamp Recordings

Whole-cell voltage clamp recordings were obtained at room temperature (~22 °C) using a HEKA EPC-10 amplifier and Pulse v8.80 (HEKA Electronic, Lambrecht, Germany) for data acquisition as previously described [27]. Electrodes were fabricated from 1.7 mm capillary glass using a Sutter P1000 micropipette puller (Sutter Instrument Company, Novato, CA, USA) and fire-polished to a resistance of 0.8–1.2 megaohms. The bath solution contained (in mM) 140 NaCl, 1 MgCl_2_, 3 KCl, 1 CaCl_2_, and 10 HEPES, adjusted to pH 7.35 with NaOH and 500 nM tetrodotoxin (TTX) to silence endogenous channels. For biophysical characterization of the Nav1.6 R639C mutant channel, we used an intracellular pipette solution that contained (in mM) 140 CsF, 10 NaCl, 1.1 EGTA, and 10 HEPES, adjusted to a pH 7.35 with NaOH. For recordings exploring the effects of CaMKII modulation on mutant channels, we used a fluoride-free internal solution to avoid complications of intracellular fluoride with calcium [38]. The internal solution contained (in mM) 10 NaMeSO_3_, 120 CsMeSO_3_, 5 EGTA, 10 HEPES, 10 Glucose, 5 MgATP, and 5 CaCl_2_, adjusted to pH 7.35 with CsOH [27]. All solutions were adjusted to 300 mOsm. To promote Ca^2+^-dependent acute activation of endogenous CaMKII, 1 µM free Ca^2+^ (calculated using WEBMAXC) was included in the internal solution in addition to 1 µM calmodulin, which was added directly before use. CaMKII-specific effects on mutant channels were assessed by measuring biophysical properties of the channel with and without the CaMKII peptide inhibitor CN21 (10 µM) or its inactive analogue, CN21Ala (10 µM), in the pipette [27]. The CN21 family of CaMKII-inhibitory peptides is derived from an endogenous CaMKII inhibitor protein, CaMKIIN [39]. CN21 is a potent and selective peptide inhibitor of CaMKII [40]. The control peptide for CN21 was developed using Spots Peptide Arrays (CN21Ala) [36].

The addition of the CaMKII inhibitor CN21 eliminates the potential for Ca^2+^ and calmodulin effects on recorded sodium currents while ensuring that the effects measured are specific to CaMKII. All voltage protocols were started 5 min after obtaining the gigaohm seal and entering whole-cell configuration to allow for enough time for diffusion of calmodulin and the additional peptides into the cell, and to control for time-dependent effects on channel properties. The liquid junction potential for these solutions was estimated to be less than 9 mV and was not corrected for this offset. The offset potential was zeroed before cells were patched. Voltage errors were minimized using a series resistance compensation greater than 70% with passive leak currents canceled by subtraction. Cells with a seal resistance greater than 1 gigaohm were initially selected for subsequent analysis.

### 2.8. Electrophysiological Analyses

Electrophysiological data were analyzed using Pulsefit (v8.80, HEKA Electronic, Lambrecht, Germany), Microsoft Excel, and Prism (v8.2, Graphpad Software, San Diego, CA, USA). For recordings using fluoride internal solutions, transient sodium currents (I_NaT_) were measured during a series of depolarizing steps ranging from −80 mV to +60 mV for 50 ms from a holding potential of −100 mV. In recordings using fluoride-free internal solutions, I_NaT_ was measured using the same voltage-step protocol from a holding potential of −80 mV to facilitate stable seals in the presence of MeSO_3_. Persistent currents were measured at the last 5.0 ms into the depolarizing step and were chosen for analysis if each cell had a seal resistance greater than 1 GΩ and a leak current smaller than 50 pA. Transient and persistent current densities were calculated by normalizing the current at each voltage to the cell capacitance. Sodium current conductance (G_Na_) was calculated using the equation
G_Na_ = I_NaT_/(V − V_rev_),(1)
where V_rev_ is the reversal potential of Na^+^ obtained in the Pulsefit software (v8.80, HEKA Electronic) for each cell. Activation curves were generated by plotting the normalized conductance against the depolarizing potentials and fitting these values with the Boltzmann function,
G_Na/_G_max_ = 1/(1 + exp[(V_50,act_ − V)/k_act_)],(2)
where G_max_ is the maximal Na^+^ conductance, V_50,act_ is the half-maximal potential for activation, V is the depolarizing potential, and k_act_ is the slope factor. The rates of decay (τ_NaT, slow_ and τ_NaT, fast_) for I_NaT_ were obtained in Pulsefit by fitting the current traces 1.0 ms into the depolarizing step with a biexponential function.

The voltage dependence of steady-state inactivation (availability) of sodium channels was measured by holding cells for 500 ms at a range of prepulse potentials from −130 mV to +40 mV, followed by a 20 ms test pulse at 0 mV to measure channel availability. Steady-state inactivation curves were generated by plotting the normalized sodium current against the corresponding prepulse potentials and subsequently fitting these values with the Boltzmann function,
I/I_max_ = 1/(1 + exp[(V_50,inact_ − V)/k_inact_]),(3)
where I_max_ is the peak sodium current, V_50,inact_ is the potential at which half of the sodium channels are available for activation, V is the prepulse potential, and k_inact_ is the slope factor. 

Recovery from fast inactivation was measured by applying an initial depolarizing step to 0 mV to measure peak current amplitude, followed by a repolarizing step to −80 mV for increasing durations and a final 20 ms test pulse to 0 mV to measure channel availability. Time constants (τ) for recovery from fast inactivation were obtained by plotting the normalized non-inactivated sodium currents against the duration of repolarizing steps fitted with a single first order exponential function.

### 2.9. Computational Simulations

Computational modeling of Nav1.6 WT and mutant sodium currents, followed by simulation of action potential firing, was conducted in the NEURON simulation environment (v7.7.2) [41]. Sodium currents were modeled using a Markov state model as previously described [13,27] with minor modifications to partially uncouple the voltage dependence of activation and inactivation [16]. Default kinetic parameters were used to model the WT Nav1.6 channel; except for epsilon, the rate constant for O → OB, was set to 0 in all simulations to eliminate resurgent current from our analysis. Mutant Nav1.6 currents were modeled relative to the WT parameters [16]. The effects of CaMKII inhibition on mutant and WT channels were modeled relative to the modeled currents in the absence of CaMKII inhibition. Curve fitting and quantitative analysis was performed to validate that the experimental changes were successfully recapitulated in our model. 

Simulations of neuronal excitability were performed in a single-compartment Purkinje neuron model [13] and in a multicompartment cortical pyramidal neuron model [42]. Action potential firing simulations in both neuron models were performed by replacing the original sodium current of the model with our modeled Nav1.6 currents. To model a heterozygous mutant condition in both models, half of the WT Nav1.6 current was replaced by the mutant sodium current. This also allowed us to more accurately reflect the CN21-mediated functional changes observed for each channel. The neuronal morphology, channel distribution, and kinetic parameters of all other remaining channels were identical to the original models. 

### 2.10. Statistics

GraphPad Prism (v8.2, GraphPad Software, San Diego, CA, USA) was used for statistical analyses and curve fitting. All datasets were tested for outlier identification using the ROUT method in GraphPad Prism and were subsequently excluded from analysis. The Shapiro–Wilk normality test was used to confirm normal distribution of each dataset. A nonlinear least-squares minimization method was used for all fitted curves. All data points are presented as the mean ± the standard error of the mean (SEM), and *n* is the number of cells used per experiment. Statistical significance was assessed using Student’s *t* test, one- or two-way ANOVA with Dunnett or Tukey post hoc test, or Sidak’s multiple comparisons test where appropriate.

## 3. Results

### 3.1. Biophysical Characterization of CaMKII Effects on R850Q Nav1.6 Mutant Channels

Our previous findings reported that acute CaMKII inhibition produces loss-of-function effects on Nav1.6 activity [27]. We hypothesized that CaMKII inhibition may attenuate gain-of-function effects in a recurrent epilepsy mutation, R850Q. Using whole-cell voltage clamp electrophysiology, we first sought to investigate the effects of CaMKII modulation on the recurrent Nav1.6 gain-of-function epilepsy mutation R850Q (ClinVar VCV000135651) [16]. The *SCN8A* gene encodes the Nav1.6 alpha subunit and comprises approximately 2000 amino-acid residues that form a complex tertiary structure containing four homologous transmembrane domains (DI–IV), each consisting of six alpha-helical segments (S1–S6). The S1 through S4 segments comprise the voltage sensing module, and the S5–S6 segments form the re-entrant P-loop that constitutes ion selectivity and the pore of the channel [43]. The R850Q mutation is located in the S4 segment of the DII voltage-sensor (Figure 1), and substitution of this residue with an uncharged Gln has been previously shown to increase persistent current in addition to producing a hyperpolarizing shift in the voltage dependence of activation [16]. To determine if CaMKII can modulate aberrant R850Q channel activity, we obtained recordings from neuronal ND7/23 cells transiently expressing R850Q channels. ND7/23 cells are from a rat DRG/mouse N18Tg2 neuroblastoma hybridoma cell line [44], which express endogenous TTX-sensitive currents, and they were used here as they typically express recombinant Nav1.6 currents at higher levels than HEK293 cells. Our Nav1.6 constructs were modified to be TTX-resistant, allowing them to be studied in isolation in ND7/23 cells in the presence of TTX. Sodium channel accessory subunits were not co-transfected with Nav1.6, in part because ND7/23 cells express several neuronal proteins including some beta subunits [45,46], and in part to reduce the variability associated with transfecting multiple cDNA constructs. To probe for CaMKII-specific effects, the peptide inhibitor CN21 or its inactive analogue, CN21Ala, was included in the patch pipette as described previously for WT Nav1.6 [27]. Representative currents traces for R850Q treated with no peptide (left), CN21 (middle), and CN21Ala (right) are shown in Figure 2A. CaMKII inhibition reduced R850Q current density by 38% compared to no peptide (at 0 mV, −191.8 ± 20.2 for no peptide (*n* = 11 cells) vs. −115.4 ± 14.1 for CN21 (*n* = 12 cells); *p* < 0.0001) and 31% compared to the control peptide CN21Ala (−166.4 ± 19.7 for CN21Ala (*n* = 8 cells) vs. −115.4 ± 14.1 for CN21 (*n* = 12 cells); *p* = 0.0100; F (56, 811) = 1.700, *p* = 0.0014, two-way ANOVA, Tukey’s post hoc) (Figure 2B). However, no changes in persistent current amplitude were observed (F (40, 351) = 0.6103, *p* = 0.9709; two-way ANOVA, Tukey’s post hoc) (Figure 2C). Similar to the previously reported effects on WT Nav1.6 [27], R850Q displayed an activation midpoint 6.13 mV more positive with CaMKII inhibition compared to no peptide (V_50_ = −6.94 ± 0.83 mV for CN21 (*n* = 10 cells) vs. −13.07 ± 1.72 mV for no peptide (n = 11 cells), *p* = 0.0097) or the control peptide CN21Ala (V_50_ = −6.94 ± 0.83 mV for CN21 (*n* = 10 cells) vs. −12.50 ± 1.49 mV for CN21Ala (*n* = 6 cells), *p* = 0.0525; F (2, 24) = 5.881, *p* = 0.0083, one-way ANOVA, Tukey’s post hoc), while the voltage dependence of steady-state inactivation remained unchanged (F (2, 21) = 0.1997, *p* = 0.8205, one-way ANOVA, Tukey’s post hoc) (Figure 2D). Previous reports indicated that R850Q opens prematurely, as reflected by a hyperpolarized activation midpoint [16], suggesting that CaMKII inhibition can normalize premature opening of the mutant channel. We further examined the effects of CaMKII inhibition on the rate of recovery from fast inactivation and observed no changes (F (2, 22) = 0.7341, *p* = 0.4913, one-way ANOVA, Tukey’s post hoc) (Figure 2E). Notably, these data show that CaMKII inhibition of the R850Q mutant channel produces similar effects to that of the WT Nav1.6 channel, resulting in reduced current density and a right shift in the voltage-dependence of channel activation, and they suggest that CaMKII inhibition could help mitigate the impact of some gain-of-function effects imparted by epilepsy mutations. 

### 3.2. Examination of a Rare SCN8A Variant That Potentially Impacts CaMKII Phosphorylation of Nav1.6

We next sought to investigate a rare *SCN8A* variant, R639C (ClinVar VCV000461334). The four domains of Nav alpha subunits are linked by several intracellular loops (L1–L3). These linkers, as well as cytoplasmic N- and C-termini, can undergo a variety of post-translational modifications (PTMs) and contribute to the regulation of channel function [34,47]. The R639C mutation is a missense variant that resides in the first intracellular loop (L1) of Nav1.6 (Figure 1), which is a PTM hotspot for Navs [27,32,48,49,50,51,52]. This mutation was submitted to the ClinVar database by a single submitter with indication of EIEE and no supporting phenotypic data publicly available. Although R639C is identified in population databases, its allelic frequency is less than 0.001% (2/237520, gnomAD), indicating that this mutation may potentially result in a pathogenic phenotype [53]. Importantly, R639 is invariant in Nav1.6 amongst most mammalian and vertebrate species (Figure 3). For instance, although this region of L1 is roughly 85% conserved between chicken and human Nav1.6, it shows less than 50% conservation with other human isoforms (Figure 4). This degree of evolutionary conservation suggests that R639 may reside in an important region of the Nav1.6 channel contributing to its specific physiological channel function. Substitution of the positively charged Arg with an uncharged Cys residue is predicted to destabilize L1 of the channel and interfere with structural properties, potentially causing altered PTMs and protein interactions in addition to altered protein folding and/or trafficking. 

Our previous findings demonstrated that CaMKII inhibition results in a significant attenuation of Nav1.6 sodium current and determined that current density modulation of the channel by CaMKII is occurring in L1 at T642 [27]. Interestingly, the R639C mutation resides within the CaMKII phosphorylation motif, where phosphorylation at T642 contributes to Nav1.6 channel regulation, and this motif is only present within the cardiac isoform Nav1.5 and is not conserved in other mammalian neuronal Nav isoforms. Substitution of R639 with a Cys residue is predicted to disrupt the T642 phosphorylation site (replaces a basic arginine upstream (P-3) of the P0 phospho-acceptor T642), which may result in altered CaMKII modulation of Nav1.6. To investigate if R639C alters the T642 CaMKII phosphorylation site, we performed CaMKII phosphorylation assays of immobilized tiled peptide arrays (Figure 5A,B). Peptides are commonly used as surrogates for protein substrates when evaluating many kinases, including substrate recognition for CaMKII [27,37,54]. Although a phosphorylated peptide may not fully recapitulate a true in situ substrate, point mutations in the core sequence (and immediately adjacent regions) can impact phosphorylation in a similar manner for both proteins and peptides. Peptides were constructed of 15-mers spanning amino acids 631–649 that contained the R639 site and tiled skipping two amino acids, such that possible phosphorylation of this peptide could be represented within multiple peptides [27]. We introduced mutant phospho-acceptor peptides whereby a non-phosphorylatable Ala point mutation of T642 was inserted to verify CaMKII phosphorylation at this site. While the WT peptide containing T642 was phosphorylated similar to the positive control peptides AC2 and GluN2B, introducing a Cys at R639 reduced phosphorylation, similar to the T642A, which unlike S641A, approached the low level of phosphorylation seen in N-term EAAT1 (a negative control peptide) (Figure 5A,B).

One caveat of this approach is the possible introduction of disulfide crosslinking between peptides in the immobilized peptide array as a result of Cys mutagenesis at R639, thereby blocking T642 substrate access to the kinase. Enhanced disulfide binding within peptides likely explains the reduced phosphorylation seen in WT Peptide 3 and S641A Peptide 3 (Figure 5B, green bars) due to the endogenous C645 moving toward the N-terminus of the tiled peptides. To address this potential caveat, we also performed CaMKII phosphorylation assays on WT Nav1.6 and mutant R639C soluble peptides under stringent reducing conditions to limit possible disulfide linkage and maximize kinase access to the substrate (Figure 5C). This approach revealed that soluble peptides containing R639C were not phosphorylated compared to WT peptides. In the Spots array, we replaced R639 with a conserved substitution for Cys that cannot form disulfide bonds, a serine residue. Again, we observed reduced phosphorylation, consistent with R639C disrupting the critical positive charge in the canonical CaMKII recognition motif. Together, these data indicate that R639C removes CaMKII phosphorylation at T642, which may alter CaMKII modulation of the Nav1.6 channel.

### 3.3. Predicted Impact of the SCN8A R639C Mutation on Nav1.6 Channel Function and Disease Phenotype

We next used an array of independent and ensemble predictor algorithm tools to predict the impact of the Arg to Cys substitution at position 639 on Nav1.6 function and pathogenicity. Independent predictors are computational algorithms that use unique protein and amino-acid features to determine potential deleteriousness of an amino-acid substitution and include sorting intolerant from tolerant (SIFT) [54,55], polymorphism phenotyping-2 (PolyPhen-2) [56], and Mutation Assessor [57]. Ensemble predictors integrate various independent predictors into a computational algorithm to determine pathogenicity of a variant and include the combined annotation dependent depletion (CADD) score [58,59], rare exome variant ensemble learner (REVEL) [60,61], and MetaLR [62]. Multiple predictors for variant evaluation were used to eliminate algorithm bias [63]. A description of each predictor tool, their pathogenicity cutoff values, and a summary of scores are reported in Table 1. Among the independent predictors, both PolyPhen-2 and Mutation Assessor predicted probable pathogenicity with scores of 0.959 and 0.649. While SIFT predicts substitutions with scores less than 0.05 as deleterious, a score of 0.1 provides better sensitivity for detecting possibly deleterious missense variants [54]. The SIFT score for R639C was 0.06, which may suggest that this mutation is damaging to protein function. Additionally, each ensemble algorithm predicted possible pathogenicity as determined by the suggested threshold scoring for each tool. CADD uses C-score rankings that range from 1 to 99. For example, variants in the highest 1% of all scores were assigned scores ≥20 [58]. R639C falls in the 0.25% of most deleterious possible mutations with a CADD score of 26. Similarly, REVEL and MetaLR rank scores range from 0 to 1, and variants with higher scores are more likely to be pathogenic [60,62]. It is estimated that 75.4% of disease-causing mutations but only 10.9% of benign or neutral variants have scores above 0.5. The R639C variant displayed scores of 0.779 and 0.804, respectively. Thus, independent and ensemble variant predictors classify R639C as producing a possible deleterious effect on protein function that may result in aberrant channel activity and suggest a pathogenic phenotype.

We further employed these bioinformatic predictor tools to evaluate the specificity of our analysis compared to the gain-of-function *SCN8A* epilepsy mutant, R850Q [16]. In line with the previously reported aberrant biophysical effects, the independent and ensemble predictors classify the R850Q variant as producing a probable deleterious effect on channel function and indicate a likely pathogenic phenotype (Table 1). Interestingly, R850Q shares similar scores to those of R639C, further indicating that the R639C variant may indeed result in aberrant channel activity.

### 3.4. Biophysical Characterization of R639C Mutant Channel Activity

To determine if the R639C mutation impacts Nav1.6 function, we performed whole-cell voltage clamp recordings on ND7/23 cells transiently expressing the human R639C mutant Nav1.6 channel as described previously for WT [27]. Representative traces of transient and persistent currents from wildtype (WT) Nav1.6 and the R639C mutant channel elicited with a series of depolarizing steps ranging from −80 mV to +60 mV for 50 ms from a holding potential of −100 mV are shown in Figure 6A. R639C produced a peak current density that was 72% larger than WT Nav1.6 (at 0 mV; R639C, −388.0 ± 70.8 pA/pF (*n* = 10 cells) vs. WT, −225.8 ± 35.5 pA/pF (*n* = 10 cells), *p* = 0.0225; F (28, 522) = 1.603, *p* = 0.0270, two-way ANOVA, Sidak’s post hoc) (Figure 6B) with a proportional increase in the persistent current density (at 0 mV; R639C, −2.31 pA/pF (*n* = 8 cells) vs. WT, −1.17 pA/pF; (*n* = 6 cells), *p* = 0.1482; F (18, 203) = 0.8606, *p* = 0.6268, two-way ANOVA, Sidak’s post hoc) (Figure 6C). We also interrogated the effect of R639C on open-channel inactivation kinetics. The rate of decay for the transient current was reflected with a bi-exponential function; therefore, fast (τ_NaT,f_) and slow (τ_NaT,s_) time constants were estimated. However, the R639C mutation did not significantly affect either component for the rate of decay for the transient current compared to WT (τ_NaT,f_; F (4, 134) = 0.4594, *p* = 0.7654, τ_NaT,s_; F (4, 128) = 0.3146, *p* = 0.8678, unpaired Student’s *t*-test) (Figure 6D).

Next, we examined whether R639C displayed altered voltage dependence of activation and steady-state inactivation (i.e., channel availability). Activation and inactivation curves for R639C and WT Nav1.6 fit with a single Boltzmann function are shown in Figure 6E. R639C displayed an activation midpoint voltage 5.08 mV more negative than the WT Nav1.6 channel (R639C, −19.99 ± 2.07 mV (*n* = 9 cells) vs. WT, −14.91 ± 2.08 mV (*n* = 10 cells), *p* = 0.1025; *t* = 1.726, df = 17, unpaired Student’s *t*-test). Although the shift in activation was not significant, a 5 mV hyperpolarizing shift of the mutant R639C channel could indicate premature activation. Compared to WT Nav1.6, channel availability for the mutant remained unaltered (R639C, −66.10 ± 1.74 mV (*n* = 10 cells) vs. WT, −67.17 ± 1.55 mV (*n* = 9 cells), *p* = 0.6541; *t* = 0.4561, df = 17, unpaired Student’s *t*-test) and did not display a significant change in the rate of recovery from fast inactivation (R639C, 13.48 ± 1.20 ms (*n* = 10 cells) vs. WT, 12.52 ± 1.50 ms (*n* = 10 cells), *p* = 0.6226; *t* = 0.5008, df = 18, unpaired Student’s *t*-test) (Figure 6F). In total, these data are consistent with the R639C mutant producing gain-of-function effects on Nav1.6 channel activity and agree with the bioinformatic predictions suggesting aberrant channel function.

### 3.5. Biophysical Characterization of CaMKII Effects on R639C Nav1.6 Mutant Channels

As with the R850Q mutation, we investigated the effects of CaMKII modulation on the Nav1.6 gain-of-function R639C mutation using whole-cell voltage clamp electrophysiology. Representative current traces for R639C treated with no peptide (left), CN21 (middle), and CN21Ala (right) that were elicited with depolarizing voltage steps from −80 mV to +60 mV for 50 ms from a holding potential of −80 mV are shown in Figure 7A. Endogenous CaMKII inhibition with CN21 reduced R639C current density by 50% compared to no peptide (at 0 mV; −273.53 ± 33.13 for no peptide (*n* = 10 cells) vs. −137.45 ± 51.16 for CN21 (*n* = 11 cells), *p* < 0.0001) and 59% compared to CN21Ala (−334.45 ± 56.76 for CN21Ala (*n* = 11 cells) vs. −137.45 ± 51.16 for CN21 (*n* = 11 cells), *p* < 0.0001; F (56, 841) = 2.587, *p* < 0.0001, two-way ANOVA, Tukey’s post hoc) (Figure 7B) with a proportional decrease in the persistent current density with CN21 treatment (Figure 7C). CaMKII inhibition of R639C produced no significant changes in voltage-dependence of activation (F (2, 26) = 2.105, *p* = 0.1421, one-way ANOVA, Tukey’s post hoc) or channel availability (F (2, 27) = 0.7649, *p* = 0.4752, one-way ANOVA, Tukey’s post hoc) (Figure 7D), or in the recovery from fast inactivation (Kruskal–Wallis = 2.502, *p* = 0.2862) (Figure 7E). Although less than the effects observed in WT [27], R639C partially retained sensitivity to CaMKII modulation and displayed reduced current density with CaMKII inhibition. These data suggest that, while the biophysical effects of the channel due to CaMKII modulation remained consistent (i.e., reduced current densities observed for R850Q and R639C and a right shift in channel activation observed in R850Q with CaMKII inhibition), the response to this modulation varied considerably between mutants and could be indicative of alterations to L1 or the phosphorylation status of the channel.

### 3.6. Impact of Putative Epilepsy Mutations on AP Firing in Simulated Neurons

To examine how CaMKII modulation of the R639C and R850Q mutant Nav1.6 channels may impact neuronal excitability, we performed simulations of AP firing in a single-compartment model of a Purkinje neuron [13] and a multicompartment model of a cortical pyramidal neuron [42]. In order to simulate spontaneous and evoked APs in these models, we first adjusted the sodium channel parameters of a Markov state sodium channel model [13] with minor modifications (see Section 2) to reflect the functional changes observed in the mutant channels with and without CaMKII inhibition by CN21. WT sodium channel parameters were modeled as previously described [27]. The functional changes in R639C were simulated relative to the WT channel, and the R850Q parameters were modeled as reported previously [16]. Effects of CN21 were modeled as relative changes to our WT and mutant Nav1.6 currents and were modeled separately to more accurately reflect how CaMKII inhibition may affect a neuron with heterozygous R639C or R850Q expression [16,27]. Voltage-clamp simulations were performed to validate that the relative experimental changes in the mutant channels with and without CN21 were recapitulated in our model (Figure 8). A summary of the model modifications is reported in Table 2.

Simulations of spontaneous AP firing frequency in WT, R639C, and R850Q Purkinje neurons are shown in Figure 9. To represent heterozygous expression of the mutant channel, half of the Nav1.6 WT channel parameters were replaced by the modeled R639C or R850Q mutant channel. The R639C and R850Q modeled neurons demonstrated increased excitability exemplified by higher firing frequencies of both spontaneous (Figure 9A) and evoked (Figure 9B) APs. The R639C neuron spontaneously fired 67% more than the WT neuron, while the R850Q neuron almost doubled in frequency. Furthermore, both modeled mutant neurons displayed a continuous increase in evoked firing frequency. Although the R639C neuron fired APs at a frequency lower than that of R850Q, higher stimulations resulted in the R639C neuron closely resembling the firing frequency of the R850Q mutant. 

Further modeling the effects of CN21 treatment on mutant neurons revealed decreased AP firing frequencies. As previously reported, modeling the effects of CN21 on the WT neuron resulted in a significant reduction of AP firing across the entire stimulation range [27]. While the R639C + CN21 neuron revealed a 20% reduction in spontaneous AP firing compared to the R639C neuron under control conditions, it was, however, still more excitable than the WT neuron under control conditions. The R639C + CN21 neuron also displayed an intermediate pattern of evoked AP firing, with the lower stimulation range closely mimicking the WT neuron and increasing in the higher stimulation range, suggesting that CaMKII inhibition might reduce susceptibility of repetitive AP firing in the mutant neuron at lower stimulation intensities. In contrast to R639C, CN21 shifted the activation midpoint of R850Q in the positive direction in addition to decreasing the current density. Therefore, one might expect an even greater silencing effect on the R850Q neuron than the R639C neuron with CN21. Indeed, simulating the effects of CN21 on the R850Q neuron resulted in a complete ablation of spontaneous AP firing. Interestingly, the evoked AP firing frequency of the R850Q + CN21 neuron closely resembled that of the WT neuron under control conditions. These results indicate that CN21 can attenuate the gain-of-function effects produced by R850Q in modeled Purkinje neurons, while resulting in an intermediate attenuation of the effects produced by R639C.

To further investigate how CaMKII modulation of the R639C and R850Q mutations can affect neuronal excitability, we adapted a more advanced multicompartmental model of a cortical pyramidal neuron [42]. This model applies multiple sodium channel isoforms native to an adult cortical pyramidal neuron with differential channel distribution at the proximal and distal AIS, in addition to dendrites, soma, and nodes, as well as along the myelinated axon. Again, we replaced the original Nav1.6 sodium current of this model with our modeled currents and replaced half of the WT Nav1.6 currents with mutant currents to reflect heterozygous expression of each variant. As with the Purkinje model, the R639C and R850Q cortical pyramidal neurons resulted in a marked increase in AP firing frequency (Figure 10). Moreover, both mutant neurons began firing sooner than the WT neuron, reflected by premature opening due to their negative shifts in activation midpoint, and they maintained a continuous increase in firing frequency across the entire stimulation range. This marked increase in frequency might be expected to facilitate repetitive neuron firing, a key characteristic of seizure activity in epilepsy. Simulating the effects of CN21 on the WT and R639C neurons resulted in decreased excitability and required more current to induce firing. However, the WT + CN21 and R639C + CN21 neurons eventually mirrored the firing frequency of the WT and R639C neurons without CN21 in higher stimulations, again suggesting that CaMKII inhibition might be more effective at conferring resistance to repetitive AP firing at lower stimulation intensities. In contrast, the R850Q + CN21 neuron maintained a profoundly reduced AP frequency compared to the R850Q neuron and largely mirrored the phenotype of the WT neuron, once again indicating that CN21 treatment of the R850Q neuron can attenuate its gain-of-function effects and bring excitability back down to WT levels.

## 4. Conclusions

In this study, we investigated the effects of CaMKII inhibition on two distinct gain-of-function Nav1.6 mutations and showed differing responses to CaMKII modulation. We showed that CaMKII modulation of the Nav1.6 epilepsy mutation R850Q largely recapitulated effects previously reported for WT Nav1.6 [27], displaying decreased current density and a depolarized activation midpoint with CaMKII inhibition. We also examined a rare missense variant, R639C, which is located within a regulatory hotspot for CaMKII modulation of Nav1.6. Using multiple prediction algorithms, R639C was predicted to have possible pathogenic effects on Nav1.6 function and shared similar prediction scores to the recurrent *SCN8A* epilepsy mutation R850Q. Functional studies using whole-cell voltage clamp electrophysiology revealed that R639C displayed gain-of-function effects, consistent with the hypothesis that a majority of epilepsy-associated *SCN8A* mutations primarily result in enhanced channel function. Interestingly, R639C channels showed altered CaMKII phosphorylation, and, in contrast to R850Q, CaMKII inhibition decreased current density without shifting the activation midpoint. Computational simulations demonstrated that modeled neurons bearing the R639C mutation are hyperexcitable, indicating that this gain-of-function mutation may negatively impact neuronal homeostasis. Most importantly, we showed that simulating the effects of CaMKII inhibition on modeled neurons harboring these mutations reduced hyperexcitability, supporting the hypothesis that acute CaMKII inhibition may represent a promising mechanism to attenuate gain-of-function effects produced by *SCN8A* epilepsy mutations. Furthermore, due to the complex molecular and structural landscape of Nav1.6 regulation, these results suggest that substitution of a single amino acid within different regions of the channel may impact Nav function in diverse ways, including direct (structural) and indirect (regulatory) mechanisms.

Aberrant Nav1.6 activity is a major underlying cause in epilepsy. A growing number of missense mutations are being identified in *SCN8A*, the majority of which may not have available phenotypic data and remain uncharacterized. Therefore, distinguishing between pathogenic and nonpathogenic missense variants is critical for interpreting how mutations may contribute to disease. From a practical perspective, variant filtering is often performed using an absence approach, which assumes that disease-causing variants likely do not reside in population databases. In some cases, however, the allelic frequency of an uncharacterized mutation can exhibit an extremely low distribution that is highly consistent with known pathogenic variants. R639C is a missense variant with an allele frequency <0.001%, and R639, in addition to a majority of residues in L1 of Nav1.6, is evolutionarily conserved across most mammalian and vertebrate species. More importantly, L1 is a critical hotspot for channel regulation, and amino-acid substitutions in this region can profoundly impact Nav regulatory properties [27,32,34,48,49,50,51,52,67]. Substitution of R639 with a Cys may destabilize L1 and contribute to altered channel function and regulation. Despite a lack of current phenotypic data to extrapolate from, the contribution of R639C as a potential pathogenic variant cannot be discounted and warrants further characterization. Indeed, prediction algorithms used in this study suggest that R639C may be potentially linked to pathogenic channel function; however, these algorithms may underestimate the importance of specific PTMs for channel function. Notably, subsequent studies revealed substantial gain-of-function effects on channel activity and altered channel phosphorylation. The two most likely mechanisms underlying the increase in current with the R639C variant are (1) an increase in open probability and (2) an increase in membrane surface expression. Epilepsy-associated *SCN8A* channelopathies are largely heterogeneous in their pathogenicity, phenotypes, and responses to treatment [15]. Even the same mutation in different patients, e.g., R1617Q, can produce diverse effects on seizure susceptibility and treatment responsiveness [68], suggesting additional underlying molecular mechanisms may contribute toward heterogeneity of the disease. To that end, Navs display a vast and complex network of post-translational regulation through myriad signaling pathways that can significantly impact channel function and expression, even beyond what is predicted for a particular mutation. As such, a strong modulator of Nav1.6 activity is CaMKII, a multifunctional Ser/Thr protein kinase critical in regulating neuronal excitability [29]. We recently showed that CaMKII inhibition produces decreased Nav1.6 transient and persistent currents, in addition to a right shift in the voltage dependence of activation, effects that are mediated by the respective phosphorylation sites T642 and S561 in the L1 hotspot for channel PTMs [27]. Here, we reported that CaMKII can produce different modulatory responses in activity between Nav1.6 mutant channels, altering current density of R639C channels while additionally shifting the activation midpoint for R850Q channels. The R639C mutation is located in L1 at the P-3 position from T642, a key phosphorylation site that contributes to CaMKII-dependent Nav1.6 modulation. CaMKII phosphorylates multiple sites on Nav1.6, and inhibiting CaMKII produces a loss-of-function-like phenotype of channel activity [27]. While LC–electrospray ionization/MS could not resolve the specificity in this site and a potential phosphorylation site on an adjacent residue (Ser561), peptide arrays and in vitro kinase assays indicated that Thr642 was phosphorylated by CaMKII, and mutagenesis of Thr642 to a non-phosphoacceptor amino acid (i.e., Ala) reduced current density and occluded CN21 effects on mutant channels. The voltage dependence of activation and availability curves did not reveal any gating changes in the T642A mutant channel, suggesting that T642 phosphorylation by CaMKII regulates current density with no observable consequences on channel gating. R639C is predicted to ablate a critical residue in the CaMKII recognition motif (basic residue at the P-3 position) and significantly reduce phosphorylation at the downstream target residue. Indeed, T642 phosphorylation level in the R639C peptide approached that of the T642A mutation using our immobilized peptides, as also seen for the R639S mutant. Using soluble peptides where the peptides can be characterized to minimize disulfide bonding, we also observed a significant loss of CaMKII activity toward the R639C mutant. We also do not think that R639C promoted phosphorylation of an adjacent phosphorylatable site S641 according to the inability of CaMKII to phosphorylate the R639C mutant. However, in the Spots assays, the phosphorylation of the R639C mutant was above the double mutant (S641A/T642A) and above the EAAT1 negative control, which was consistent with our observation that the R639C channels remained partially sensitive to CaMKII-dependent modulation. While a reduction in the R639C current density in response to CaMKII inhibition was observed, we saw no CaMKII-dependent shift in the voltage dependence of activation. Thus, it is possible that R639C may allow limited phosphorylation of T642 or altered the phosphorylation at S561 (which is important for CaMKII-dependent effects on channel activation [27]), promoted CaMKII-dependent phosphorylation at a different site in Nav1.6, or influenced other regulatory signaling pathways to impact channel function.

Differences in CaMKII modulation may also be due in part to the location of Nav mutations in relation to sites of regulation. Whereas R639C is located in close proximity to known CaMKII phosphorylation sites in Nav1.6, R850Q resides within a transmembrane segment displaced from intracellularly accessible CaMKII phosphorylation motifs. Therefore, in contrast to R639C, we predicted that the functional effects of CaMKII inhibition on the R850Q mutant channel would mirror the functional effects observed for WT Nav1.6. Indeed, CaMKII inhibition of R850Q significantly reduced transient current density and shifted the activation midpoint to the right. Previous biophysical characterization of R850Q revealed gain-of-function effects on channel activity, including a left shift in the activation midpoint and increased persistent currents [16]. Thus, our results indicate that CaMKII inhibition can attenuate specific gain-of-function effects produced by R850Q. However, unlike WT Nav1.6, CaMKII modulation of R850Q did not affect persistent current generation. This indicates that pathological persistent currents generated by mutation may be driven by different mechanisms than the physiological Nav1.6 persistent current modulated by CaMKII. To this end, reports have shown that pathological persistent currents generated by diverse *SCN8A* epilepsy mutations can be differentially targeted pharmacologically. For example, cannabidiol can effectively reduce persistent currents generated by the *SCN8A* epilepsy mutant N1768D, but has no effect on mutant channel L1331V persistent currents [69]. These results suggest that differing modulatory responses between *SCN8A* mutants may arise from differences in the amino-acid substitution and/or regulatory properties of the channel.

To predict how CaMKII modulation of mutant sodium channels may impact excitability at the single-neuron level, we performed computational simulations using Purkinje and cortical pyramidal neuron models [13,42]. We demonstrated that heterozygous expression of R639C in both models increased the firing frequency and produced hyperexcitability, suggesting that this gain-of-function mutation may potentially contribute to increased seizure susceptibility. More importantly, we showed that modeling CaMKII inhibition in simulated mutant neurons could attenuate hyperexcitability associated with mutation. Simulating CaMKII inhibition on the R850Q neuron reduced excitability to a greater extent than the R639C neuron, resulting in firing that largely resembled the WT neuron in both models. These differences were likely due to CaMKII inhibition only targeting R639C current density without affecting channel activation, whereas CaMKII inhibition of R850Q produced an additional right shift in the voltage-dependence of activation, thus delaying channel opening compared to R639C. Because CaMKII is believed to be dynamically regulated by Ca^2+^ spike frequency [29,70], future studies may reveal whether different CaMKII activation states may differentially impact substrate selectivity [37] or if graded levels of CaMKII activation differentially modulate WT versus mutant Nav1.6 channels to impact neuronal activity. Studies in primary neurons have provided novel insight with other *SCN8A* variants [71] and, thus, could help further our understanding of how CaMKII modulates Nav1.6 currents in distinct neuronal populations.

Given its pivotal role in neuronal signaling, it is not surprising that mutations in Nav1.6 associated with gain-of-function phenotypes promote hyperexcitability in *SCN8A* channelopathies. Therefore, modulating neuronal excitability may be achieved by targeting Nav1.6 to decrease channel function. Interestingly, dozens of *SCN8A* mutations reported in ClinVar are located in the PTM hotspot L1 which displays a high degree of evolutionary conservation. Despite its importance in regulating channel activity and expression, a vast majority of mutations in L1 remain uncharacterized. Amino-acid substitutions within this highly conserved region can disrupt intramolecular interactions with the channel in addition to PTMs, thus profoundly impacting channel function and contributing to pathogenic changes associated with excitability disorders, like epilepsy. Our study demonstrated that R639C, a Nav1.6 mutation that interferes with post-translational regulation of the channel, impacted Nav function. We also showed that diverse Nav1.6 mutants displayed distinct responses to CaMKII modulation and revealed that CaMKII inhibition may attenuate gain-of-function effects commonly observed in epilepsy-associated *SCN8A* mutations. Taken together, these data further highlight the importance of CaMKII modulation on Nav1.6 activity and may influence how specific mutations might be therapeutically targeted.

## Figures and Tables

**Figure 1 cells-11-02108-f001:**
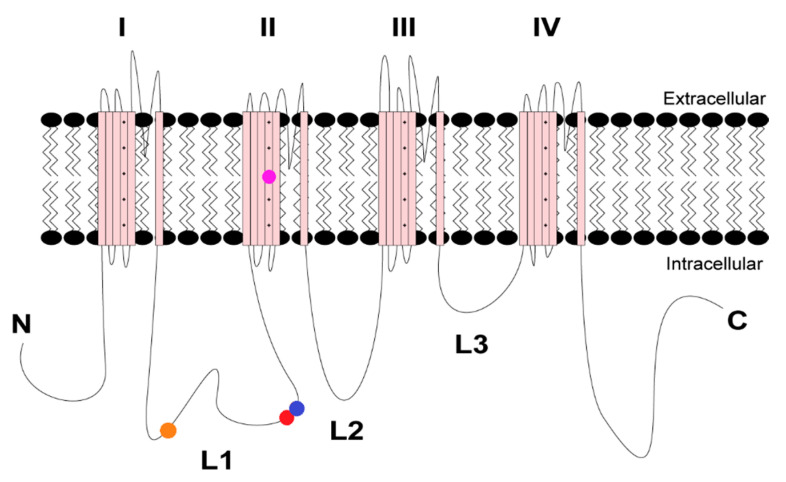
Topology of voltage-gated sodium channel *SCN8A*. The R639C and R850Q mutations are denoted in red and fuchsia, respectively. The CaMKII phosphorylation sites S561 and T642 in L1 of the channel are in orange and blue, respectively. The four pseudo-domains are indicated with the Roman numerals I–IV and the three large intracellular loops with L1–L3.

**Figure 2 cells-11-02108-f002:**
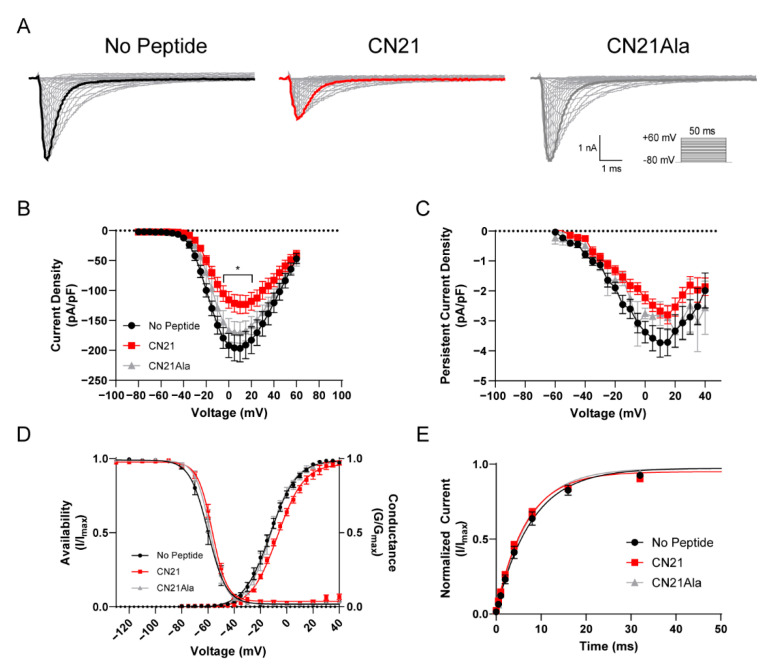
CaMKII inhibition of the R850Q mutant channel reduces the current density and shifts the activation midpoint to more depolarizing potentials. (**A**) Representative family of current traces generated by Nav1.6 mutant R850Q with no peptide (black, left), CN21 (red, middle), or CN21Ala (gray, right) in the patch pipette elicited with a voltage–step protocol (inset). Peak current traces are in boldface. (**B**) Transient current density–voltage and (**C**) persistent current density–voltage curves. (**D**) Voltage dependence of activation and availability curves for R850Q treated with or without CN21 or CN21Ala each fitted with a Boltzmann function. (**E**) Rate of recovery from fast inactivation. CaMKII inhibition reduces R850Q current density and shifts the voltage dependence of activation to more positive potentials. * *p* < 0.05; *n* = 5–12 per group; two-way ANOVA ± SEM (error bars), Tukey’s post hoc test.

**Figure 3 cells-11-02108-f003:**
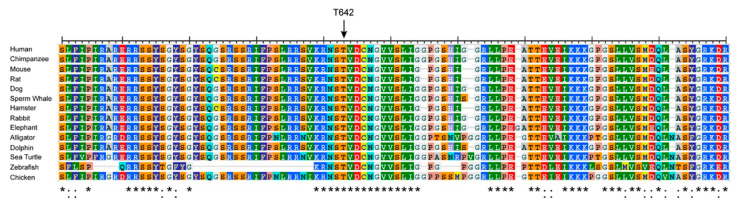
Nav1.6 protein sequence alignments of L1 region spanning amino acids 600–696 (human *SCN8A*) compared to other Nav1.6 orthologs. T642 is denoted by the arrow. An * (asterisk) indicates a fully conserved amino acid residue. A: (colon) indicates conversation of amino acid residues with similar properties.

**Figure 4 cells-11-02108-f004:**
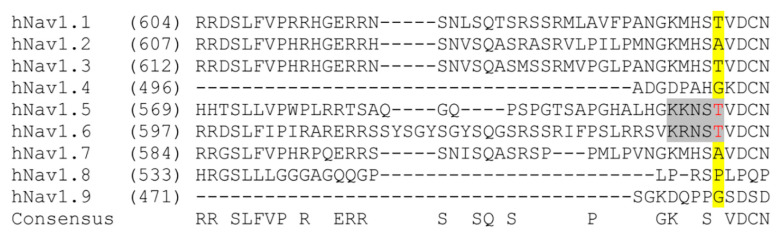
Nav1.6 protein sequence alignment spanning position 642 (yellow) in L1 region of the channel. The T642 site (red) within the CaMKII phosphorylation motif (gray) is only present in human isoforms Nav1.5 and Nav1.6.

**Figure 5 cells-11-02108-f005:**
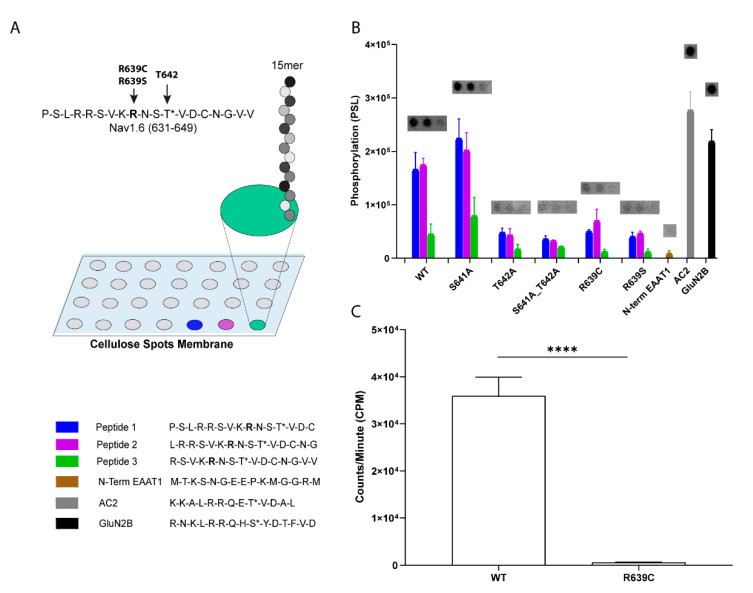
R639C reduces CaMKII-dependent phosphorylation of key modulatory site, T642. (**A**) Schematic of the Spots immobilized peptide tiling assay performed in (**B**). (**B**) Average phosphorylation intensity of immobilized WT peptides tiled from amino acids spanning 631–649 (Peptides 1–3) after αCaMKII phosphorylation in the presence of [γ-^32^P]-ATP. Mutant peptides included Ala substitution for Thr/Ser, and Cys or Ser replacement for R639. (**C**) Average phosphorylation intensity of soluble peptides WT (RSVKRNSTVDCNGVV) and R639C (RSVKCNSTVDCNGVV) after αCaMKII phosphorylation in the presence of [γ-^32^P]-ATP. R639C (as well as R639S) ablates phosphorylation at T642 in both immobilized and soluble kinase assays. The N-terminus of EAAT1 served as the negative control; AC2 and GluN2B phosphorylation served as positive controls. **** *p* < 0.0001; *n* = 3. Student’s *t*-test ± SEM (error bars). The * (asterisks) in the key (bottom left panel) indicate a CaMKII phosphorylation site.

**Figure 6 cells-11-02108-f006:**
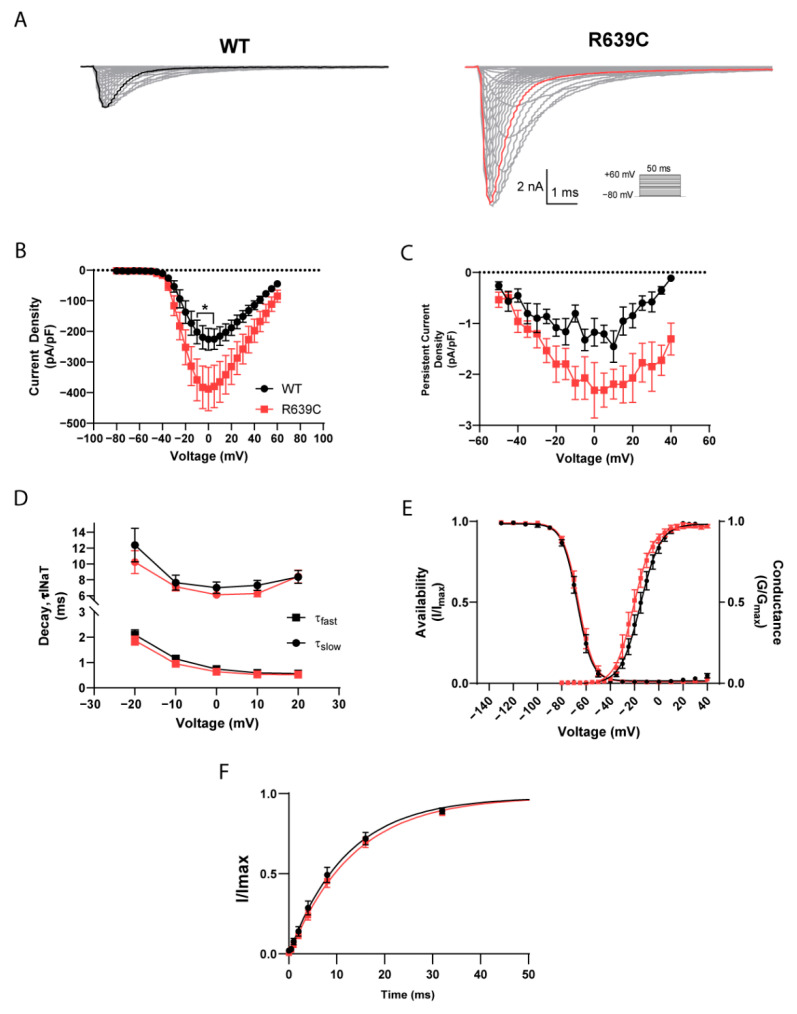
R639C displays substantially increased current density. (**A**) Representative current trace families from WT Nav1.6 (left) and R639C (right) channels elicited by a voltage-step protocol (inset). Boldface traces represent peak current density. (**B**) Transient and (**C**) persistent current density–voltage curves for WT Nav1.6 and R639C. (**D**) Rate of decay for transient sodium current (squares, fast τNaT; circles, slow τNaT) across a range of depolarization voltages. (**E**) Voltage dependence of activation and availability curves, each fitted with a Boltzmann function. (**F**) Rate of recovery from fast inactivation. R639C displays increased transient and persistent current densities compared to WT Nav1.6 in addition to a 5.08 mV left shift in the activation midpoint. * *p* < 0.05; *n* = 6–10 per group; two-way ANOVA ± SEM (error bars), Sidak’s post hoc test; Student’s *t*-test was performed on parameters extrapolated from curve fitting.

**Figure 7 cells-11-02108-f007:**
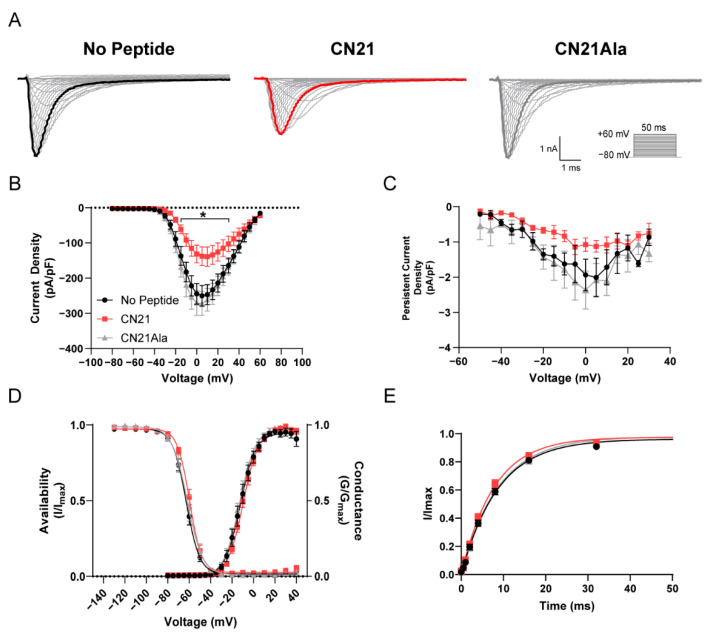
CaMKII inhibition decreases R639C mutant current density. (**A**) Representative family of current traces generated by Nav1.6 mutant R639C with no peptide (black, left), CN21 (red, middle), or CN21Ala (gray, right) in the patch pipette elicited with a voltage-step protocol (inset). Peak current traces are in boldface. (**B**) Transient current density–voltage and (**C**) persistent current density–voltage curves. (**D**) Voltage dependence of activation and availability curves, each fitted with a Boltzmann function. (**E**) Rate of recovery from fast inactivation. CaMKII inhibition reduces R639C current density with no additional changes in biophysical properties. * *p* < 0.05; *n* = 4–11 per group; two-way ANOVA ± SEM (error bars), Tukey’s post hoc test.

**Figure 8 cells-11-02108-f008:**
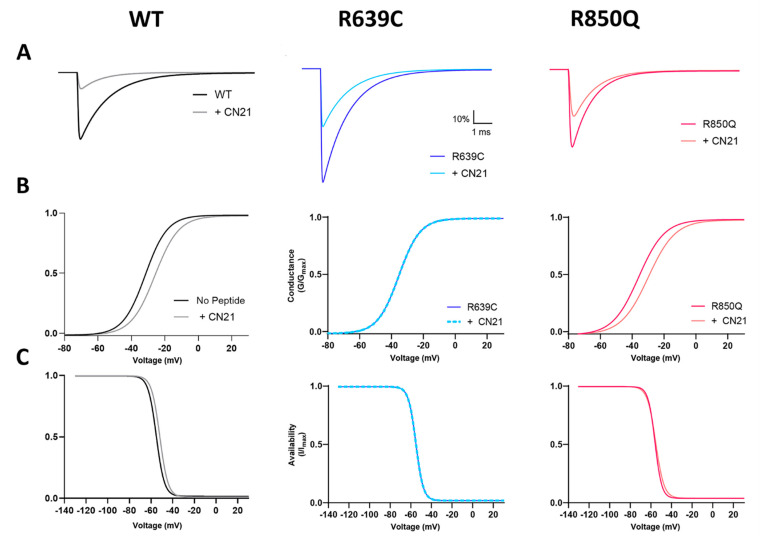
Voltage-clamp simulations of modeled WT and mutant Nav1.6 channels with and without CN21-mediated functional alterations. (**A**) Current traces for WT (black), R639C (dark blue), and R850Q (red) mutant Nav1.6 generated from modeled neurons without open-channel block. Effects of CaMKII inhibition with CN21 were modeled as well and are shown in gray for WT + CN21, cyan for R639C + CN21, and pink for R850Q + CN21. (**B**) Voltage dependence of activation and (**C**) steady-state inactivation curves of the modeled WT and mutant Nav1.6 channels with and without CN21-mediated functional alterations.

**Figure 9 cells-11-02108-f009:**
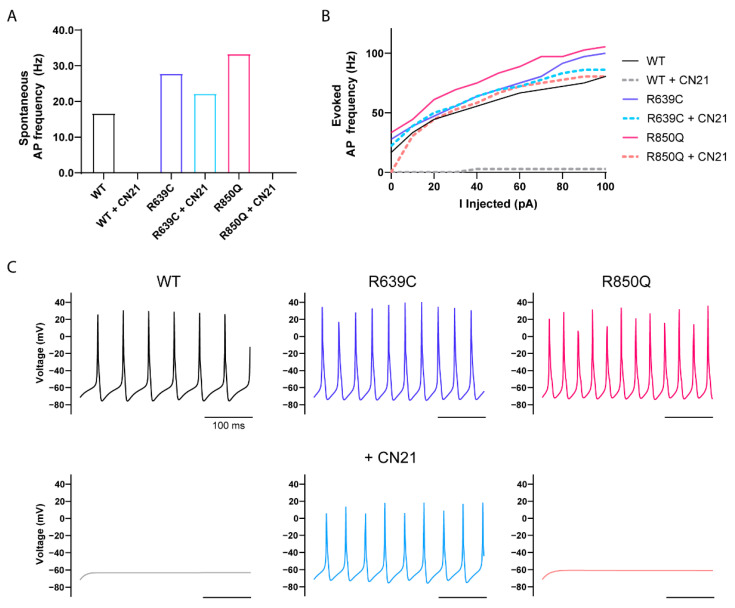
Simulations of action potential firing in a Purkinje neuron expressing heterozygous mutant Nav1.6 channels with and without CaMKII inhibition. Quantification of spontaneous (**A**) and evoked (**B**) action potential frequency from modeled Purkinje neurons expressing either 100% WT Nav1.6 or 50% WT and 50% mutant Nav1.6 channels to reflect heterozygous expression (WT, black; WT + CN21, gray; R639C, blue; R639C + CN21, cyan; R850Q, red; R850Q + CN21, pink). Representative traces of spontaneous action potentials are shown in (**C**).

**Figure 10 cells-11-02108-f010:**
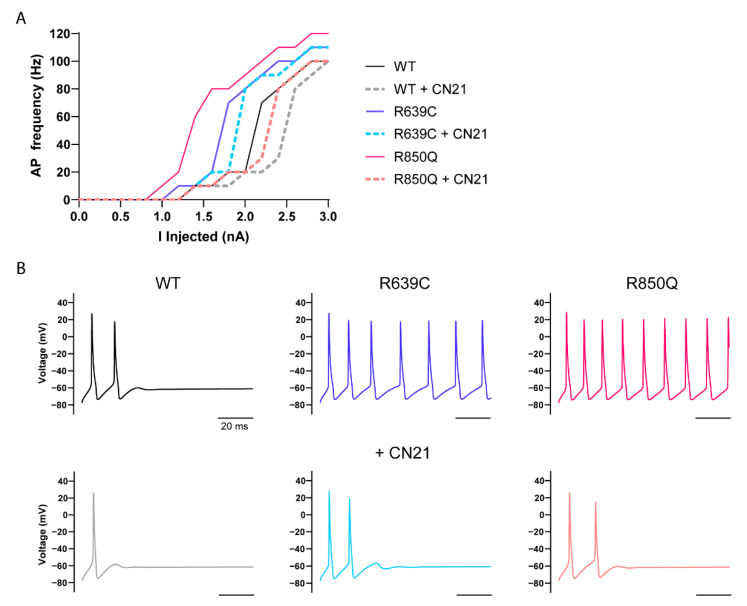
Simulations of action potential firing in a cortical pyramidal neuron expressing heterozygous mutant Nav1.6 channels with and without CaMKII inhibition. (**A**) Quantification of evoked action potential frequency from modeled cortical pyramidal neurons expressing either 100% WT Nav1.6 or 50% WT and 50% mutant Nav1.6 channels to reflect heterozygous expression (WT, black; WT + CN21, gray; R639C, blue; R639C + CN21, cyan; R850Q, red; R850Q + CN21, pink). Representative traces of action potentials evoked by 1.8 nA current injection are shown in (**B**).

**Table 1 cells-11-02108-t001:** Pathogenicity predictions for mutant Nav1.6 channels. Brief description of prediction algorithm tools used to predict deleteriousness/pathogenicity of mutant *SCN8A* channels and includes scoring information for each predictor. The *SCN8A* transcript LRG1389 was used as a reference.

Predictor	Description	Pathogenicity Cutoff	R639C	R850Q
**SIFT**	Predicts whether an amino-acid substitution will have a phenotypic effect on protein function using sequence homology information and the physical properties of amino acids [54,64].	<0.1	0.06	0.0
**PolyPhen-2**	Uses sequence homology, phylogenetic, and structural information to predict the impact of an amino-acid substitution on the structure and function of a human protein [56,65].	>0.446	0.959	0.999
**Mutation Assessor**	Predicts the functional impact of an amino-acid substitution in proteins using evolutionary considerations, including sequence homology and 3D structure, in protein homologs. Scores range from 0 to 1, with high scores reflecting greater likelihood of deleteriousness [57,66].	>0.5	0.649	0.994
**CADD**	Predicts pathogenicity of a variant with C-scores by integrating diverse genomic features, including conservational, epigenetic, and functional considerations [58,59].	>20	26	33
**REVEL**	Predicts pathogenicity of a missense variant using an ensemble of individual predictor tools. Scores range from 0 to 1 and variants with higher scores are more likely to be pathogenic [60].	>0.5	0.779	0.987
**MetaLR**	Integration of independent variant deleteriousness scores and allele frequency information to predict pathogenicity of missense variants. Scores range from 0 to 1 and variants with higher scores are more likely to be pathogenic [62].	>0.5	0.804	0.992

**Table 2 cells-11-02108-t002:** Parameters of modeled sodium channels.

	WT	WT + CN21	R639C	R639C + CN21	R850Q	R850Q + CN21
**Density**	Default	0.300×	1.72×	0.878×	1.23×	0.835×
**C_off_**	0.5	0.5	0.25	0.25	0.4	0.4
**O_on_**	0.75	1.3	0.75	0.75	1.018	1.018
**O_off_**	0.005	0.005	0.007	0.007	0.018	0.018
**ε**	0	0	0	0	0	0
**α**	150	92	277.5	277.5	85	50.3
**β**	3	3	3.7	3.7	0.9	0.9
**α_i_**	150	150	110	110	126	250
**β_i_**	3	3	4	4	3.4	4
**γ**	150	150	130	130	137	137

## Data Availability

The data that support the findings of this study are available upon request from the authors.

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
