# Peer review of "CaMKII Inhibition Attenuates Distinct Gain-of-Function Effects Produced by Mutant Nav1.6 Channels and Reduces Neuronal Excitability"

_cells, 2022, doi:10.3390/cells11132108_

Round 1
Reviewer 1 Report
This manuscript extends the authors previous work on how post-translational modifications of the Nav1.6 voltage-gated sodium channel alter channel function, with an emphasis on CAMKII signaling and epilepsy-causing mutations.
Major points
1. The voltage clamp is performed in ND cells which may not have the same posttranslational modifications as found in real cortical or hippocampal neurons. Do the authors have any data from, for example, cultured hippocampal neurons? At the very least the use of ND cells should be justified.
2. What of the beta subunits? Where they co-transfected or are Beta 1 and 2
available in the ND cells?
3. What is the likely mechanism underlying the increased current density of R639C relative to the WT, i.e. does this mutation primarily enhance channel surface expression?
4. The Fig 6 legend title mentions a hyperpolarized shift in activation but the text states that this shift is not significant. Thus it seems the voltage-dependence of activation should not be in the legend title.
5. Some discussion of why peptide phosphorylation should be expected to reflect what occurs in the intact protein is needed.
6. More background should be provided, e.g. on the use of the CN21 peptides and what we already know about the functional role of T642 phosphorylation.
Minor points
1. Persistent current was measured at what time?
2. Are the channels GFP-tagged or is soluble GFP co-transfected?
3. Perhaps add panels with WT currents to Figs 2 and 7.
4. Yellow is a bad color choice in Fig 1. Also, the circles denoting the various amino acid locations could be enlarged.
Reviewer 2 Report
In this manuscript, the authors examined the effects of CaMKII inhibition on two distinct gain-of function Nav1.6 mutations associated (R850Q) or potentially associated (R639C) with epilepsy. The authors report different effects of the mutations, and more importantly, differing responses to CaMKII modulation. Additionally, computational simulations demonstrated that modeled neurons bearing the mutations are hyperexcitable, and that acute CaMKII inhibition may represent a promising mechanism to attenuate these commonly observed gain-of-function effects associated with Nav1.6 mutations. This is an interesting study that provides further insights into the effects of a previously uncharacterized Nav1.6 channel mutation in epilepsy (R639C) on channel function and neuronal excitability, and into the possibility of attenuating these Nav1.6 channel gain-of-function effects using CaMKII inhibition. However, in order to strengthen their arguments and better support the conclusions made, there are a number of points that the authors need to address.
Major comments:
1- Evaluation of the persistent Na+ current in Figures 2C, 6C and 7C should be performed as a percentage of the peak Na+ current. This is especially important here because changes in peak Na+ current density are observed. These novel data analyses will strengthen the authors arguments, so it will then be possible to conclude on whether or not CaMKII inhibition reverses the effects of the mutation(s) on the persistent Na+ current or whether the observed changes on persistent Na+ current are proportional/subsequent to the changes in peak Na+ current density. As demonstrated in their previous study (Zybura AS et al, J Biol Chem 2020), the authors showed that CaMKII inhibition decreases the persistent Na+ current generated by WT Nav1.6 channels, whereas no inhibition are observed with both of the mutants studied here. These novel data analyses would therefore strengthen the authors arguments that the R639C mutation does not regulate the persistent Na+ current and that there is no reversal, whereas the R850Q mutation increases the persistent Na+ current, which is not reversed either.
2- Examination of the voltage-dependence of activation properties of Na+ channels is not trivial and could easily be biased by technical limitations, particularly changes in the peak Na+ current density. As a consequence, to strengthen their arguments, it seems necessary to increase the n values of recorded cells, which are currently small (n=4-12 cells per group). In Figure 6E, for example, the authors conclude that the R639C mutant “could” have premature activation (line 457) although the shift in voltage-dependence of activation is not significant. What should we conclude? Could the non-significant shift be artefactually caused by the increased peak Na+ current density? The same flaw applies to the results presented in Figure 7D in which no effects on voltage-dependence of activation are observed, whereas possible changes associated with CaMKII inhibition could be expected.
3- In the voltage-clamp simulations of mutant Nav1.6 channels with or without CaMKII inhibition (Figures 8A-C), it would be nice and helpful for the reader to include the simulation of the WT Nav1.6 channel with or without CaMKII inhibition even if this information was published before (Zybura AS et al, J Biol Chem 2020). In order to distinguish these previous and newest data, maybe the WT simulation could be presented using dotted lines. In addition, it would seem relevant to include the changes in persistent Na+ current in these voltage-clamp simulations. Also, are they included in the analyses of the action firing properties (Figures 9 and 10)?
Minor comments:
- A table recapitulating all the voltage-clamp data (current densities, biophysical properties, n values, statistical analyses) would help evaluating and recapitulating the conclusions made.
Reviewer 3 Report
Voltage-gated sodium channels have a crucial role with regard to neuronal function, and mutations in sodium channel genes are responsible for genetic epilepsy syndromes with a wide range of severity. To date, many genetic variants have been described in sodium channel genes, however, their functional impact on protein function and cellular behavior often remains elusive. A better understanding of these underlying mechanisms might open new roads for pharmacological interference.
In their present manuscript, which is based on a set of experimental approaches as well as simulations, Zybura and colleagues concentrate on two variants within the Scn8a gene; the R639C mutation and their recently described R850Q mutation (Pan and Cummins 2020). The authors perform electrophysiological recordings in transiently transfected ND7/23 cells and found a gain-of-function effect for R639C mutant channel activity. In addition, they claim that the R639C mutation ablates CaMKII phosphorylation at the T642 site. The manuscript raises a few concerns and questions as follows:
Major points:
- The authors state in their manuscript that R639 removes CaMKII phosphorylation at T642, which may alter CaMKII modulation of the Nav1.6 channel. This statement is based on Figure 5. However, Figure 5 asks for some additional clarifications:
Why did the authors not show the phosphorylation state of T642A itself (without R639 mutation)? In their previous paper (PMID: 32611770, Figure 7), they performed similar experiments, where they showed that T642 by itself already strongly reduced the phosphorylation state of the corresponding peptides. Is the effect they observed for R639C caused by the mutation, or also already by the WT variant? The authors should show all corresponding controls for this experiment. The authors should also perform a statistical test to see whether real differences exist between the different groups before they can draw any conclusions (e.g. is R639C significantly different from
R639_S641A?), and the authors should test for differences within the groups (e.g. is the third bar of WT significantly different from the first bar? And the first bar from AC2 different from 2-3?). Are the three bars, the three different peptides? It would also help the reader to better interpret the results if the authors would show the SPOTs for all peptides.
- The authors perform their experiments in ND7/23 cells and use simulations in Purkinje neurons and pyramidal neurons, comparable with the experiments described in Pan and Cummins 2020. Their conclusions would definitely benefit from parallel experiments in “real-life” primary neuronal cultures, instead of modeling experiments (e.g. Liu et al., 2019. PMID: 30615093).
Minor points:
- The legends should be improved. For example, the legends of Figure 5B and 5C are intermingled. The same for Figure 6E and 6F.
- The authors should include significance levels also within their figures (e.g. Figure 5C)
Round 2
Reviewer 1 Report
My concerns have been addressed.
Reviewer 2 Report
I accept the replies from the authors and would like to remove my limitations for publication.
Reviewer 3 Report
...